# Effectiveness and Therapeutic Mechanism of Pharmacopuncture for Pain in Parkinson’s Disease: A Study Protocol for a Pilot Pragmatic Randomized, Assessor-Blinded, Usual Care-Controlled, Three-Arm Parallel Trial

**DOI:** 10.3390/ijerph20031776

**Published:** 2023-01-18

**Authors:** Jung-Hee Jang, Jieun Kim, Ojin Kwon, So Young Jung, Hye-Jin Lee, Seung-Yeon Cho, Jung-Mi Park, Chang-Nam Ko, Seong-Uk Park, Hyungjun Kim

**Affiliations:** 1Clinical Medicine Division, Korea Institute of Oriental Medicine, Daejeon 34054, Republic of Korea; 2Clinical Research Coordinating Team, Korea Institute of Oriental Medicine, Daejeon 34054, Republic of Korea; 3Department of Korean Medicine Cardiology and Neurology, Graduate School, Kyung Hee University, Seoul 02447, Republic of Korea; 4Department of Cardiology and Neurology, College of Korean Medicine, Kyung Hee University, 26, Kyungheedae-ro, Dongdaemun-gu, Seoul 02447, Republic of Korea

**Keywords:** acupuncture therapy, Parkinson’s disease, nociceptive pain, neuroimaging

## Abstract

Pain in Parkinson’s disease (PD) represents a complex phenotype known to decrease quality of life. This pragmatic randomized, controlled clinical trial evaluated the efficacy of pharmacopuncture (PA) for improving pain symptoms and investigated the corresponding therapeutic mechanisms in patients with PD. Ninety patients with PD-related pain were randomly allocated to receive either PA, manual acupuncture, or usual care in a 1:1:1 ratio; sixty healthy controls were included for comparative analysis of brain imaging data. Over 12 weeks, study treatment provided 2 days per week for 8 weeks with a follow-up period of 4 weeks. The primary outcome measure was the King’s Parkinson’s Disease Pain Scale score for assessing improvement in PD-related pain, including a sub-analysis to investigate the pattern of changes in pain according to a PD-related pain mechanism-based classification. Secondary outcome measures included a numerical rating scale-based assessment of the intensity and location of pain and changes in pain-associated symptoms, such as depression, anxiety, and sleep disorders. Exploratory outcome measures included structural and functional brain patterns on magnetic resonance imaging, blood molecular signature changes, gait analysis, facial expression and movement assessment in response to emotional stimuli, and a traditional Korean medicine syndrome differentiation questionnaire. The trial findings provided important clinical evidence for the effectiveness of PA in the management of PD-related pain and its associated symptoms, and helped elucidate the mechanism of its therapeutic effect on PD-related pain.

## 1. Introduction

The incidence of Parkinson’s disease (PD) is increasing owing to the aging of society. Pain as a non-motor symptom of PD is experienced by a vast majority of patients, with an incidence rate of 40–86%. Furthermore, pain has a remarkable impact on the quality of life of patients with PD, but treatment guidelines often do not take pain into account [1]. The etiology and phenotype of pain in PD are complex and multifactorial [2]; it is classified into five types according to Ford’s scheme—musculoskeletal, radicular-neuropathic, dystonic, central neuropathic, and akathisia pain [3]. Recently, Mylius et al. reported a PD pain classification system with three pain types based on the mechanism of chronic pain: nociceptive pain, such as that in musculoskeletal pain syndromes; neuropathic pain such as radicular or central pain; and nociplastic pain, where the nociceptive system is overactive even without any tissue damage or lesions in the somatosensory system [4]. Most patients with PD experience nociceptive pain such as musculoskeletal pain due to abnormal postures and visceral pain due to abnormal functioning of the vegetative nervous system. Pain management in PD is focused on optimizing dopaminergic treatment related to insufficient dopaminergic supply or dopamine-sensitive central pain. However, pain management therapy is only effective in approximately 30% of PD patients [5]. Therefore, it is necessary to establish the pathophysiological mechanisms and pathways underlying pain in PD using specific evaluation scales that consider the pain type. Moreover, as patients with PD also present with anxiety and depression [6,7], changes in pain-related symptoms in patients with PD also need to be investigated.

Pharmacopuncture (PA) is a traditional medical therapy that involves injecting herbal extracts into acupoints based on pharmacology and meridian theory, and combining herbal medicine and acupuncture to achieve synergistic effects. *Jungsongouhyul* PA is one of the eight principal PAs based on the traditional medicine syndrome differentiation (SD) of *Yin-Yang*, exterior-interior, hot-cold, and deficiency-excess [8]. Specifically, *Jungsongouhyul* PA prescribes extravasated blood treatment involving the herbal drugs *Gardeniae Fructus*, *Corydalis Tuber*, *Olibanum*, *Myrrha*, *Persicae Semen*, *Paeoniae Radix Rubra*, *Salviae Miltiorrhizae Radix*, and *Sappan Lignum* for adjusting the function of the body and improving pathological conditions [9]. It is used to treat various conditions including pain, neurological diseases, and gastrointestinal diseases [10], and is most often applied to manage musculoskeletal disease [11]. Combination therapy that includes *Jungsongouhyul* PA has been found to be demonstrably beneficial for patients with acute traumatic shoulder pain [12] and patients with cervical pain caused by traffic injury [13]; moreover, crush-induced sciatic nerve injury in a rat model was used to show that the neuroprotective effect of combination therapy including *Jungsongouhyul* PA was related to pain reduction and motor nerve recovery, decreased substance-P expression, and increased brain-derived neurotrophic factor expression [14,15]. As such, PA is used to treat pain symptoms and for nerve recovery. The pain in PD is mostly caused by musculoskeletal pain activated by mechanical, thermal, or mechanical stimuli related to nonneural tissues and to increased muscle rigidity. PD pain is also classified as neuropathic pain associated to a lesion or disease of the peripheral or central somatosensory systems [4]. Therefore, PA will help relieve musculoskeletal pain and neuropathic pain in PD. PA has the advantages of a more rapid effect, easier dosage adjustment, and additional synergistic effects from the acupuncture and herbal medicine extracts [11]. Furthermore, PA as a non-pharmacological approach is useful for addressing patient reluctance in terms of increasing pill burden, fear of drug interactions, and poor tolerability or response to analgesics [15]. Therefore, *Jungsongouhyul* PA may be a promising modality for pain treatment in PD.

However, limited clinical trials have assessed the efficacy of *Jungsongouhyul* PA for pain in patients with PD and the corresponding therapeutic mechanisms. Hence, it is difficult to confirm whether *Jungsongouhyul* PA is effective for pain treatment in PD. Recently, SU *Eohyeol* PA with enhanced efficacy was developed by adding *Cervi Pantotrichum Cornu* to *Jungsongouhyul* PA. Herein, we present a study design and trial protocol to evaluate the effectiveness of SU *Eohyeol* PA for pain management in patients with PD. We also aim to elucidate the therapeutic mechanism underlying the effectiveness of PA in patients with PD by analyzing the levels of molecular signatures in the blood and the structural and functional brain patterns on magnetic resonance imaging (MRI).

## 2. Materials and Methods

### 2.1. Aims

To assess the effectiveness and safety of PA for directly improving PD-related pain and pain-associated motor and non-motor symptoms.To investigate, in an exploratory manner, the possibility of neuroimaging and molecular signature indicators as biomarkers of the therapeutic response to PA for PD.To explore the pathogenesis of pain subtypes in PD, as well as the corresponding current therapy guidelines for traditional medicine, such as traditional Korean medicine.

### 2.2. Study Design and Setting

This clinical study was a pragmatic randomized, usual care (UC)-controlled, three-arm trial to investigate the effectiveness of PA for the treatment of pain in patients with PD. PD patients who meet the eligibility criteria will be enrolled and randomly allocated to the treatment group (PA with conventional therapy), manual acupuncture (MA) control group (MA with conventional therapy), or UC control group (UC with conventional therapy). The subjects underwent PA or MA 2 days per week for 8 weeks and were followed up for 4 weeks after treatment. All assessments were performed in the dopaminergic “on” state in patients with PD (at the same time, in each patient after the last session of conventional treatment). The dopaminergic “on” state refers to periods during which symptoms remain adequately controlled in response to dopaminergic medications. A flowchart describing the study design is presented in Figure 1.

The effectiveness of PA was assessed according to improvements in pain and related symptoms, changes in structural and functional brain data, and molecular signature indicators in blood samples. Previous studies have reported that structural and functional network topology on MRI differs between PD patients and healthy controls [16]. Therefore, healthy individuals were enrolled to undergo MRI assessment, and investigated the structural and functional MRI differences between PD and healthy controls. The study setting and site of data collection was the Clinical Trial Center of Kyung Hee University Hospital in Gangdong, South Korea. The study was designed and reported in accordance with the Consolidated Standards of Reporting Trials guidelines and the Standard Protocol Items: Recommendations for Interventional Trials (SPIRIT) checklist (Additional File 1). The protocol was reviewed, revised, and supported by experts on traditional Korean medicine, MRI, statistics, acupuncture, and neurologic disorders in South Korea.

### 2.3. Participant Recruitment

A total of 90 PD patients with pain and 60 healthy participants were recruited by displaying posters in the hospital and community, the internet, local newspapers, and subway advertisements from May 2022 to November 2024. Participants who met the eligibility criteria (Table 1) and provided written informed consent for study participation received information regarding the clinical study during consultation with the coordinator. Subsequently, all PD patients were randomly allocated to the PA, MA, or UC control groups. Healthy participants were individuals with no functional or organ-related diseases or clinically significant findings determined based on their medical history and physical examination.

### 2.4. Randomization

The 90 patients were allocated to the PA, MA, or UC groups in a 1:1:1 ratio by an independent statistician using computer-generated three- or six-block-size randomization. Each participant was selected via systematic sampling and was assigned an identification code enclosed in an opaque envelope. The assessors were blinded to the primary outcome and the Unified PD Rating Scale (UPDRS) II and III scores. In other words, the assessors collecting the data were blinded to the group allocation. The outcomes were evaluated by two assessors, one of whom was blinded and one who was not blinded. The enrollment of participants, allocation sequence, assignment to groups, and assessor blinding were the responsibility of the Clinical Trial Center of Kyung Hee University Hospital.

### 2.5. Interventions

All patients with PD underwent PA or MA therapy twice a week for 8 weeks. Conventional treatments, including L-DOPA, catechol-O-methyl transferase inhibitor, monoamine oxidase inhibitor, and dopamine agonist treatments, were allowed during the trial. For combination drugs, any drugs taken four weeks before participation in the study were allowed at the discretion of the researcher if they were not prohibited drugs (steroids, immunosuppressants, neuropsychiatric drugs) or other drugs that might have affected the results of the study. Analgesic drugs could be continuously taken if the dose had been stable for more than two weeks at the time of screening, but the details of drug administration (name of the drug, purpose of administration, dosage, and duration of administration) were recorded at each visit. In addition, any oriental medical therapy to improve PD pain, other than the intervention being performed in this study, was prohibited. Drugs that could affect symptoms during the period of clinical research were prohibited from being administered at the judgment of the researchers. Procedures, including details of needling (i.e., acupoints centered on the area where the participant complains of pain, needle stimulation, and retention time) to be administered in the PA and MA groups, were performed according to the clinical judgment of a senior Korean medicine doctor with more than 10 years of clinical experience based on the patient’s condition in the pragmatic randomized controlled trial (RCT). All procedural methods were retrospectively recorded. Treatment was discontinued based on the withdrawal of consent by participants or their legal representatives, violation of the eligibility criteria during the clinical trial, occurrence of adverse events (AEs), or poor compliance defined as receiving less than five of the sixteen total treatments.

### 2.6. Treatment Group: PA

The treatment group received SU Eohyeol PA that included the following nine dried medicinal herbs: *Cervi Pantotrichum Cornu* and Jungsongouhyul PA herbs (*Gardeniae Fructus*, *Corydalis Tuber*, *Olibanum*, *Myrrha*, *Persicae Semen*, *Paeoniae Radix Rubra*, *Salviae Miltiorrhizae Radix*, and *Sappan Lignum*). The medicinal herbs were extracted, distilled, and filtered for injection into the acupoints. The final form of the medicinal herb filtrate liquid used for the PA was supplied by the pharmaceutical manufacturer of Namsangcheon Korean Medicine Clinic (Republic of Korea) and was manufactured in compliance with Good Manufacturing Practices in accordance with the Ministry of Food and Drug Safety guidelines.

### 2.7. Control Group: MA or UC

The MA and UC control groups received MA or UC to improve pain and related symptoms in PD.

### 2.8. Outcome Measures

The primary outcome measure was the King’s Parkinson’s Disease Pain Scale (KPPS) score at visits 1 and 18 to evaluate the effectiveness of PA for pain in PD. Pain in PD has been previously divided into three types—PD-directly related pain (related to disease onset), PD-indirectly related pain (aggravated by the disease), and PD-unrelated pain [17]. The KPPS has been validated for PD-directly related pain. As subjects with KPPS score >0 were enrolled in this study, the effectiveness of PA was limited to PD-directly related pain. The secondary outcome measures were the numerical rating scale (NRS) score for pain area and questionnaire survey (UPDRS II and III, Beck Depression Inventory II, Beck Anxiety Inventory, Euro-Quality of Life-5 Dimension, Pain Catastrophizing Scale, and PD Sleep Scale-2) results for related symptoms. To investigate the biological mechanisms underlying PD, exploratory outcome measures included MRI findings, molecular signatures in the blood, SD questionnaire survey, gait analysis, and facial expression analysis based on facial muscle response to emotional stimulation. The schedule for outcome measure assessments is presented in Table 2.

#### 2.8.1. Primary Outcome: Assessment Scales for PD Pain

The KPPS is a self-reported questionnaire that assesses pain in PD. Fourteen items in seven domains encompass recognized categories of pain in PD: domain 1, musculoskeletal pain; 2, chronic body pain; 3, fluctuation-related pain; 4, nocturnal pain; 5, orofacial pain; 6, discoloration/edema, swelling; and 7, radicular pain. Each item is scored by multiplying the scores of the intensity (0–3) by frequency (0–4), and the total score ranges between 0–144. The KPPS is a reliable and valid scale for grading various types of pain in PD [18]. In addition, the cut-off points for distinction of pain severity levels (no pain, mild, moderate, and severe pain) in PD have been reported for the KPPS [19]. In this study, the total KPPS score, including all seven domains, was measured to assess the improvement of direct PD-related pain.

#### 2.8.2. Secondary Outcomes: Evaluation Instruments of Pain and Related Symptoms

An NRS was used to assess the intensity and location of pain symptoms in each body part. Furthermore, given that pain is reportedly associated with negative emotions in PD [6,7], pain-related symptoms including depression, anxiety, and sleep disorders were assessed using questionnaire surveys, such as the Beck Depression Inventory II [20,21], Beck Anxiety Inventory [22,23], Pain Catastrophizing Scale [24], and PD Sleep Scale-2. Pain in PD can greatly reduce patient quality of life, and thus, the change in quality of life according to pain modulation was assessed using the Euro-Quality of Life-5 Dimension [25]. In addition, changes in motor and non-motor symptoms of PD due to pain modulation were evaluated using the UPDRS II and III.

### 2.9. Exploratory Outcomes

#### 2.9.1. Investigating Therapeutic Mechanisms and Biomarkers for PA to Treat PD Pain: Molecular Analysis and Neuroimaging Using MRI

The understanding of the neural basis of pain has increased with the development of neuroimaging techniques. Neuroimaging measurements using MRI could support the understanding of the neurophysiology of pain in patients with PD. In this study, we used a 3.0T MRI scanner (3T Philips, Achieva, whole-body MRI scanner) with a head coil to acquire structural and functional brain data. MRI scans included anatomical T1-weighted imaging, T2*-weighted echo-planar imaging sequence scans obtained by diffusion tensor imaging (DTI), and resting-state functional MRI (rs-fMRI). Regional gray matter volume and cortical thickness were assessed using T1-weighted images, and white matter microstructure changes were analyzed using DTI data. Resting-state fMRI data were used to calculate estimates of functional connectivity using independent component analysis and the seed-based correlation approach.

Additionally, an investigation of changes in blood molecular markers could help us to understand the mechanism of action of PA on pain in patients with PD. The levels of transcriptomic alterations and methylation; inflammation-related cytokines such as interleukin (IL)-1β, IL-2, IL-4, IL-6, and IL-10; tumor necrosis factor-α; interferon-gamma; cortisol; and C-reactive protein were analyzed. Blood was collected from the subjects and centrifuged at temperatures below 4 °C. The obtained plasma and serum samples were stored at temperatures below −80 °C until further analysis. The samples were analyzed at the Korea Institute of Oriental Medicine.

#### 2.9.2. Investigation of the Distribution of SD for PD Pain and SD Changes following PA: SD Questionnaire Survey

A personalized approach in complementary and alternative medicine, including traditional medicine, is suitable for the treatment of PD pain, which can be categorized into several types [26]. In a previous study, the most frequent symptom patterns of PD were *Yin* deficiency of the kidney and liver, deficiency of *Qi* and blood, phlegm heat and wind stirring, blood stasis and wind stirring, and deficiency of *Yin* and *Yang* [27]. However, the distribution of SD in PD patients with pain symptoms has not been reported. Therefore, we intended to identify the pathogenesis and present a complementary and alternative medicine treatment guide by investigating cold-heat and deficiency-excess SD for pain in PD. The cold-heat and deficiency-excess SD questionnaire was sufficiently verified through item reliability evaluation, correlation coefficient analysis, and factor analysis in healthy participants in South Korea [28,29]. The findings can be a basic study in the standardizing of SD for PD pain.

#### 2.9.3. Objective Evaluation of PA for PD Pain-Related Symptoms: Gait Analysis and Assessment of Facial Expressions in Response to Emotion-Eliciting Stimuli

To assess behavioral alternations and emotional dysregulation due to pain, we tested facial expressions in response to emotion-eliciting stimuli. Facial movements were recorded using a webcam, while participants watched video clips designed to elicit specific emotional valence (i.e., joyful, sad, appetite-inducing, anxiety-inducing, and neutral). The experimental protocol was implemented based on procedures from previous studies [30,31,32]. The iMotions software (version 7.0; iMotions Inc., Boston, MA, USA) was used to quantify the probabilistic scores for 17 facial action units, which were extracted from the video recordings of participants’ faces. Facial expressions were defined as a combination of facial action units, and we evaluated the “emotional contagion” of the feelings of others.

Increased frequency of pain in PD is associated with increased motor fluctuations and freezing of gait and rigidity [33]. Therefore, changes in gait analysis according to pain modulation were measured using the objective gait parameter wearable devices G-walk and Beflex. G-walk (BTS Bioengineering S.p.A., Garbagnate Milanese, Italy) is a system that wirelessly measures and analyzes the subjects’ walking and functional performance using acceleration, gyro sensors, and magneto sensors. Velcro is used to fix the device to the height of the fifth lumbar spine, and the subject is allowed to walk in this state. Gait is evaluated using various methods with the seven built-in protocols, and information such as acceleration, rotation (angle), and direction of patient movement is collected as three-axis data. In addition, the stance/swing phase is classified by detecting the increase or decrease in acceleration according to heel-ground contact, and force values such as the take-off force and propulsion are calculated using the patient’s weight and acceleration. The parameters are wirelessly transmitted through Bluetooth to the BTS G-Studio software for analysis. Beflex (B-Lab system; Beflex, Daejeon, Taejon-jikhalsi, South Korea) is a wearable earphone device that can perform professional gait analysis only using motion information from the patient’s head, and can estimate the ground reaction force and obtain gait factor data with high accuracy. It enables the analysis of indicators such as walking cadence, head angle, stride width, vertical amplitude, left and right amplitudes, left and right balance, impact, maximum force, leg stiffness, and consistency.

### 2.10. Safety Assessment and AEs

A clinical laboratory examination was performed at baseline, before treatment, and 8 weeks after completing treatment (visit 18) as a safety assessment for all participants. All AEs, defined as undesirable and/or unexpected medical findings that were absent prior to commencing this clinical trial, were recorded. AEs were reported to the institutional review board (IRB) of Kyung Hee University Hospital at Gangdong, which determined whether the trial should be discontinued. The causal link between AEs and the procedure was categorized into six stages as determined by the research clinician: Definitely related, Probably related, Possibly related, Probably not related, Definitely not related, and Unknown. Appropriate treatment and follow-up were provided for severe AEs (SAEs).

### 2.11. Statistical Methods

#### 2.11.1. Sample Size Calculation

This study was a pilot clinical trial to determine the degree of symptomatic improvement that can be achieved with PA treatment in PD patients. Because of the lack of difference between the PA treatment group and the conventional treatment control group, the effect size was assumed based on similar previous studies, such as MA and bee venom acupuncture [34,35,36]. Accordingly, with a significance level of 5% and 80% power, the effect size was assumed to be 1.21. Considering a dropout rate, the calculated target sample size was 30 subjects per group.

#### 2.11.2. Statistical Analysis

A complete set analysis was used to analyze the effectiveness of the intervention, and the per-protocol analysis was verified. For efficacy analysis, the mixed-effects model for repeated measures method was used, with each group and visit time as fixed factors and subjects as random factors. Student’s paired t-test or Wilcoxon signed rank test were used to compare factors before and after treatment within groups. All AEs, including SAEs, were classified by study group. All statistical analyses were performed using SAS^®^ version 9.4 (SAS Institute. Inc., Cary, NC, USA), and all *p*-values were two-sided. Statistical significance was set at a 5% level.

### 2.12. Data Management and Monitoring

The investigator directly filled out electronic subject case report forms based on the survey questionnaire. The subjects were provided sufficient time to complete the questionnaires with the investigator’s assistance. All documents obtained in the clinical trial are to be preserved in the confidential archives at Kyung Hee University Hospital, Gangdong for 3 years. No information was shared without permission from the principal investigator. Regular monitoring of patient safety, investigation of AEs, and quality control of the data were performed by an independent monitoring committee unrelated to this study.

## 3. Discussion

This study investigated the clinical effectiveness and therapeutic mechanisms of PA as a non-pharmacological therapy in patients with PD pain. Pain in PD is a multifactorial phenotype [2] and can be categorized into several types [3,4]. To our knowledge, no clinical study has investigated the effect of PA on PD-directly related pain. Therefore, the primary outcome was the total KPPS score as an indicator of pain directly related to PD, as it measures the frequency, intensity, and location of each type of PD pain and rates the pain modifications associated with motor fluctuations in PD [1]. In a study on four factors of the KPPS, domains 1, 4, 6, and 7 measured skeletal/muscular pain caused by disorders in the peripheral nervous system, domain 2 measured the body’s internal pain, domain 3 assessed the pain associated with fluctuation states in PD, and domain 5 was the fourth factor [18]. By applying the domains of the KPPS to PD pain classification reported by Mylius et al. [4], domains 1, 3, 4, and 6 were included in the assessment of nociceptive pain. Domain 2 included nociplastic pain and domain 7 included neuropathic pain. Therefore, through a sub-analysis using the KPPS domains, PD pain mechanism-based classification, such as nociceptive pain, was possible. Additionally, the NRS score was a secondary outcome to evaluate pain intensity in each body part with pain.

In general, pain is a complex sensory and emotional experience that encompasses multiple somatic and affective domains. Negative affect, anxiety, and depression are commonly reported in various pain disorders [37,38,39], and patients with PD are also known to present with anxiety and depression in addition to pain [6,7]. Given the complexity of pain, we will evaluate both the sensory dimension (scales for pain intensity) and affective dimension (scales for depression, anxiety, and catastrophizing) of pain and their association in patients with PD.

Numerous neuroimaging studies on chronic pain have provided evidence on brain function and the networks supporting the neural mechanisms of somatic and affective pain processing. Several neuroimaging studies in patients with PD have demonstrated a correlation between aberrant functional connectivity of the dopaminergic pain pathway region and pain intensity [40], poorer pain network topology than that in healthy controls [16], and increased connectivity of the serotonergic brain region to pain-related brain regions [41]. A previous study revealed that persistent pain in PD is associated with gray matter changes and functional brain networks [42]. However, the neural mechanism underlying the association between the somatic and affective domains of pain in PD remains unclear. In our study, we adopted multimodal brain imaging techniques and multidimensional pain evaluations. Based on previous studies, we investigated changes in pain-related symptoms and the structural and functional differences between patients with PD and healthy controls. Furthermore, the association between the central mechanism of pain processing will be investigated using brain estimates and multidimensional pain scales.

One limitation of this study is the fact that it was a small pilot clinical trial. However, this study could potentially be a basis for future large-scale clinical studies, thereby addressing the lack of evidence on the effectiveness of PA in the treatment of pain in PD. Another limitation is that this study could set the conditions of the interventions; the control group was administered UC instead of a placebo because of the pragmatic RCT design, and the efficacy of PA as an intervention could not be confirmed. However, this pragmatic RCT reflecting standard therapy and UC may provide results that can directly inform real-world clinical practice [43]. In addition, the usefulness of sham acupuncture as a placebo control method remains debatable because its beneficial effects cannot be excluded [44]. Moreover, masking is not possible for sham acupuncture, given the nature of the intervention [43]. Consequently, the control group in this trial received MA or UC. Further research will be required to validate our findings.

## 4. Ethics and Dissemination

The IRB of Kyung Hee University Korean Medicine Hospital, Gangdong approved the protocol of this clinical trial and accepted responsibility for supervising all aspects of the study (approval number: KHNMCOH 2022-02-005-001). Study participation was voluntary, and a signed informed consent form was obtained from all participants. All participants and their families were informed of the potential risks and benefits of treatment. The study protocol (version 1.2, 7 March 2022) was registered with the Clinical Research Information Service, the 11th member to have joined the World Health Organization International Clinical Trials Registry Platform as the primary registry representing the Republic of Korea (registration number: KCT0007254). In cases of significant protocol changes, the research coordinator requested additional approval from the IRB and trial registries. The results of this study were disseminated through publications in peer-reviewed journals.

## 5. Conclusions

To the best of our knowledge, this study was the first clinical trial on the clinical effectiveness of PA, which is commonly used in South Korea to treat pain in PD, and to include neuroimaging changes and molecular signature indicators. We expect to demonstrate improvements in pain and related symptoms, and changes in the levels of molecular markers. Our data provide insights into the mechanisms underlying the therapeutic effects of PA and helps to identify potential complementary therapies for pain in PD.

## Figures and Tables

**Figure 1 ijerph-20-01776-f001:**
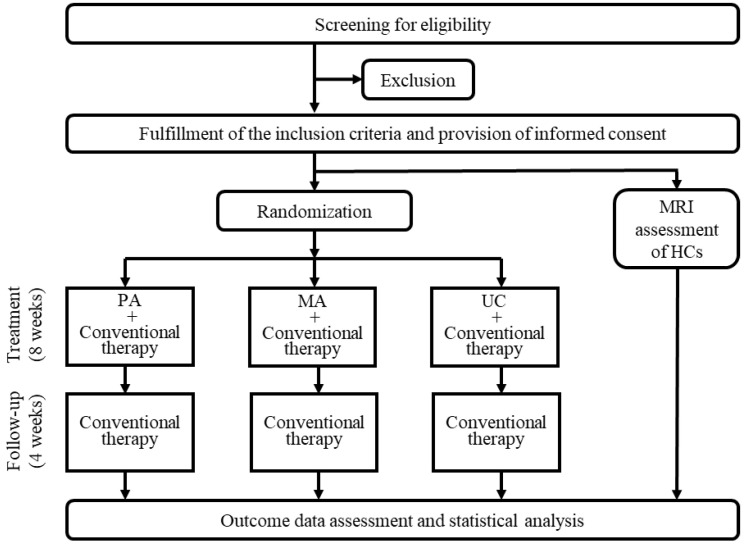
Subject selection flowchart. PA, pharmacopuncture; MA, manual acupuncture; UC, usual care; MRI, magnetic resonance imaging; HC, healthy control.

**Table 1 ijerph-20-01776-t001:** Patient eligibility criteria.

Inclusion Criteria
Adult patients (age ≥ 19 years)
Diagnosis of idiopathic PD based on the UK Parkinson’s Disease Society Brain Bank criteria
Hoehn and Yahr Scale stage 1–4
King’s Parkinson’s Disease Pain Scale score > 0
A stable dose of conventional treatment for at least 4 weeks prior to enrollment
Terminated PA or MA treatment 4 weeks before enrollment
Provide voluntary written informed consent to participate in this clinical study
**Exclusion criteria**
Parkinson-plus syndromes (i.e., multisystem atrophy, progressive supranuclear palsy, corticobasal degeneration, and dementia with Lewy bodies)
Pain unrelated to PD (e.g., postoperative pain)
History of neuropsychiatric disorder unrelated to PD
Moderate or higher cognitive impairment that will interfere with the evaluation of treatment effects
Severe acute cardiovascular disease (i.e., heart failure, myocardial infarction, stroke, and hypertension)
Serious conditions (i.e., anemia, active pulmonary tuberculosis, thyroid disease, and other infectious and systemic diseases)
Active cancer
Uncontrolled hypertension (systolic blood pressure > 160 mmHg or diastolic blood pressure > 100 mmHg)
History of hypersensitivity reactions to PA
Indications for PA treatment being inappropriate or unsafe (i.e., hemorrhagic disease, patients with severe diabetes who have a higher risk of infection)
History of taking oral adrenal corticosteroids (steroids), immunosuppressants, or antipsychotic drugs or other drugs that may affect clinical trial results within the last 4 weeks
Inability to undergo MRI
Pregnant or lactating women or current contraceptive use among women of pregnant potential who are likely to become pregnant (except for women who have undergone sterilization)
History of drug or alcohol abuse
Unstable medical condition as determined by the research clinician; A patient who shows clinically significant diseases and disorders in physical or clinical examination or is receiving active treatment thereof
Participation in another clinical trial within the last 4 weeks
History of vaccination within 4 weeks or plans to be vaccinated during the clinical trial period
Inappropriate for enrollment due to other reasons as determined by the investigator

PD, Parkinson’s disease; PA, pharmacopuncture; MA, manual acupuncture; MRI, magnetic resonance imaging.

**Table 2 ijerph-20-01776-t002:** Schedule of enrollment, clinical outcomes, and safety assessment.

		Screening	MRI	Treatment			MRI	Follow-Up
Week		−2 to −1	−1 to −0	1	4	8	8 to 9	12
Visit No.		1	2	3 (Baseline)	10	18	19	20
Provide informed consent		O						
Identification of inclusion and exclusion criteria		O						
Demographics		O						
Medical and disease history		O						
Vital signs		O	O	O (every visit)	O (every visit)	O	O	O
Clinical laboratory examination		O				O		
Identification of eligibility criteria for MRI measurement		O						
Randomization		O						
Outcome measures	KPPS	O		O	O	O		O
	NRS			O	O	O		O
	UPDRS II, III			O	O	O		O
	Pain catastrophizing score			O		O		
	EQ-5D			O		O		
	PDSS-2			O		O		
	Facial expression analysis			O		O		
	Syndrome differentiation		O				O	
	BDI II		O				O	
	BAI		O				O	
	Gait analysis			O		O		O
MRI measurement			O				O	
Blood collection for molecular analysis		O				O		
PA or MA or UC				O (every visit)	O (every visit)	O		
Treatment compliance						O		
Identification of concomitant drug change		O	O	O (every visit)	O (every visit)	O	O	O
Identification of adverse reaction(s)					O (every visit)	O	O	O

MRI, magnetic resonance imaging; KPPS, King’s Parkinson’s Disease Pain Scale; NRS, numerical rating scale; UPDRS, Unified Parkinson’s Disease Rating Scale; EQ-5D, Euro-Quality of Life-5 Dimension; PDSS, Parkinson’s Disease Sleep Scale; BDI, Beck Depression Inventory; BAI, Beck Anxiety Inventory; PA, pharmacopuncture; MA, manual acupuncture; UC, usual care

## Data Availability

The source document data for the clinical trials were deposited in the Clinical Trial Center of Kyung Hee University Hospital at Gangdong repository after completion of this study (KHNMCOH 2022-02-005-001).

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
