# Peer review of "Effectiveness and Therapeutic Mechanism of Pharmacopuncture for Pain in Parkinson’s Disease: A Study Protocol for a Pilot Pragmatic Randomized, Assessor-Blinded, Usual Care-Controlled, Three-Arm Parallel Trial"

_ijerph, 2023, doi:10.3390/ijerph20031776_

Round 1

Reviewer 1 Report

Please give some examples or write it specifically about the following phrase. "Unstable medical condition as determined by the research clinician" 

Author Response

RE: ijerph-2068722, Effectiveness and therapeutic mechanism of pharmacopuncture for pain in Parkinson's disease: a study protocol for a pilot pragmatic randomized, assessor-blinded, usual care-controlled, three-arm parallel trial

Dear Editor and Reviewers,

We appreciate the Reviewers' helpful comments and suggestions. We have added more detailed information to address the issues raised and provided further explanations. In addition, the overall English grammar was corrected, and some contents were modified according to the journal guidelines. The responses to the Reviewers' comments are presented below.

Response to the Reviewer #1's Comments

Comments:

Please give some examples or write it specifically about the following phrase. "Unstable medical condition as determined by the research clinician"

Response

We would like to thank the reviewer for the comment. We have mentioned this point in section 2.3 of the revised manuscript as follows: "Unstable medical condition as determined by the research clinician. A patient who shows clinically significant diseases and disorders in physical or clinical examination or is receiving active treatment thereof."

Reviewer 2 Report

This is a very ambitious and comprehensive study, for which I would like to congratulate the authors.

My concerns:

1. With so much current global instability, I am concerned that recruitment may be more difficult than anticipated, although more than 2 years have been allocated for the process. Wide-ranging exclusion criteria may also affect recruitment - (Covid) vaccination may be an issue, for example.  

2.1. Is 12 weeks sufficient? Eight weeks of treatment may provide benefit for some patients; should they then be offered further treatment following their involvement in the study?

2.2. No mention is made of longer-term follow-up - eg. at 6 months after the initial 12-week period. This could be important and useful.

3. Use of sample size calculations and safety assessments is commendable., as is the exclusion of sham acupuncture.

4. I find Figure 2 confusing. Are the authors really going to analyse all the data they gather? If not, should so much information be collected? Prioritise?

5. The "six stages" on p. 9 should be explained (and referenced).

6. The "dopaminergic “on” state" should be explained.

7. Use of t-tests should be justified. Will data be checked for normality of distribution? Should non-parametric methods be used instead?

8. "Procedures, including details of needling (i.e., acupoints, needle stimulation, and retention time) to be administered in the PA and MA groups will be performed according to the clinical judgment of a senior Korean medicine doctor with more than 10 years of clinical experience based on the patient’s condition in the pragmatic randomized controlled trial (RCT)."

This tells me nothing! Many readers may know very little about pharmacopuncture, or even acupuncture. It would be good to know more, particularly about the points to be used, and how they will be selected. For instance, will they be general body points, or points local to painful areas?

What model will be used - "traditional medicine syndrome differentiation" is a very general description. 

How will points be selected? According only to traditional theory, or based on the results of previous published research as well? And will the herbs be adjusted according to syndrome as well? For some patients, the combination chosen may be less appropriate than for others. 

Will it be possible to analyse results for subgroups of patients with different syndromes or severity/type of PD? Results may be excellent for one, poor for another. Useful findings may be missed without subgroup analysis - although I realise this may mean that more patients need to be recruited.   

10 years of clinical experience may mean very little actual experience with PD, unless the senior Korean medicine doctor who is allocating treatment has been working in a neurology setting. I would be happier if I knew more about this senior Korean medicine doctor, and would prefer it if there were case discussions to plan treatment rather than depending on one person's view.

Author Response

RE: ijerph-2068722, Effectiveness and therapeutic mechanism of pharmacopuncture for pain in Parkinson's disease: a study protocol for a pilot pragmatic randomized, assessor-blinded, usual care-controlled, three-arm parallel trial

Dear Editor and Reviewers,

We appreciate the Reviewers' helpful comments and suggestions. We have added more detailed information to address the issues raised and provided further explanations. In addition, the overall English grammar was corrected, and some contents were modified according to the journal guidelines. The responses to the Reviewers' comments are presented below.

Response to the Reviewer #2's Comments

This is a very ambitious and comprehensive study, for which I would like to congratulate the authors.

Thank you for your positive comment.

Comments:

  1. With so much current global instability, I am concerned that recruitment may be more difficult than anticipated, although more than 2 years have been allocated for the process. Wide-ranging exclusion criteria may also affect recruitment - (Covid) vaccination may be an issue, for example.

Response

Pain caused by the COVID-19 vaccine can bias the research results; therefore we have excluded it. We also think that recruiting such participants will be difficult. As an alternative, we plan to actively promote advertisements for participant recruitment. In addition, in the case of vaccinated persons, enrollment after a certain period of time is considered.

2.1. Is 12 weeks sufficient? Eight weeks of treatment may provide benefit for some patients; should they then be offered further treatment following their involvement in the study?

Response

The duration of treatment was determined after sufficient discussion with clinicians with extensive clinical experience in pharmacopuncture or acupuncture treatment for pain in Parkinson's disease. There is no plan to provide further treatment in this study.

2.2. No mention is made of longer-term follow-up - eg. at 6 months after the initial 12-week period. This could be important and useful.

Response

We also think that longer-term follow-up is essential, especially because there are existing cases showing differences between the experimental and control groups after a long period of acupuncture research. However, this study is a pilot study to evaluate the effectiveness of pharmacopuncture for pain in Parkinson's disease. Based on the results of this study, we will consider extending the treatment period and follow-up period in follow-up studies.

In previous study on “Acupuncture Effect and Mechanism for Treating Pain in Patients With Parkinson’s Disease”1, acupuncture treatment consisted of one to three sessions per week that were separated by at least 1 day and lasted for 8 weeks. And patients with PD who received acupuncture treatment showed improvement in KPPS. Although it is a study on the effect of acupuncture rather than pharmacopuncture, it can serve as a basis for the intervention period of this study.

Reference

1        Yu, S. W. et al. Acupuncture Effect and Mechanism for Treating Pain in Patients With Parkinson's Disease. Front Neurol 10, 1114, doi:10.3389/fneur.2019.01114 (2019).

  1. Use of sample size calculations and safety assessments is commendable., as is the exclusion of sham acupuncture.

Response

Thank you for your comment.

  1. I find Figure 2 confusing. Are the authors really going to analyse all the data they gather? If not, should so much information be collected? Prioritise?

Response

We plan to analyze the data of all outcome measures, and experts for clinical indicators, neuroimaging, and blood analysis are involved in this study. After analyzing each part, we plan to analyze the correlation with clinical index changes, including primary outcome.

In addition, since there are many outcomes in this study, it may be burdensome to the participants. To solve this problem, we calculated the time required to assess the outcome index. Finally, we decided that the evaluation sessions were taken for two days (e.g., baseline evaluation is divided into visits 1 and 2). This decision was made after having sufficient discussions with a clinical senior Korean doctor who routinely treats patients with PD. Although fig 2 may seem confusing, we think that even elderly patients with Parkinson's disease can be included in this study and assessed correctly.

  1. The "six stages" on p. 9 should be explained (and referenced).

Response

We have mentioned this point in the 2.10 section of the revised manuscript as follows: "The causal link between AEs and the procedure will be categorized into six stages as determined by the research clinician: Definitely related, Probably related, Possibly related, Probably not related, Definitely not related, and Unknown"

  1. The "dopaminergic “on” state" should be explained.

Response

Because dopaminergic drugs have a wearing off effect, "on" state differs for each patient; therefore, a specific time cannot be specified after taking the conventional treatment. We have set it to a time when the participant is free from discomfort due to PD's symptoms and can visit the hospital. And we want to set it to the same time for each patient after the last session of conventional treatment.

We have mentioned this point in section 2.2 of the revised manuscript as follows: " The dopaminergic “on” state refers to periods during which symptoms remain adequately controlled in response to dopaminergic medications."

  1. Use of t-tests should be justified. Will data be checked for normality of distribution? Should non-parametric methods be used instead?

Response

We have mentioned this point in section 2.11.2 of the revised manuscript as follows: "Student’s paired t-test or Wilcoxon signed rank test will be used to compare factors before and after treatment within groups.

  1. "Procedures, including details of needling (i.e., acupoints, needle stimulation, and retention time) to be administered in the PA and MA groups will be performed according to the clinical judgment of a senior Korean medicine doctor with more than 10 years of clinical experience based on the patient’s condition in the pragmatic randomized controlled trial (RCT)."

This tells me nothing! Many readers may know very little about pharmacopuncture, or even acupuncture. It would be good to know more, particularly about the points to be used, and how they will be selected. For instance, will they be general body points, or points local to painful areas?

Response

We have mentioned this point in section 2.5 of the revised manuscript as follows: “acupoints centered on the area where the participant complains of pain”.

Pragmatic randomized controlled trial (pRCT) design focusing on effectiveness rather than efficacy that best reflects the likely clinical response in practice.2 Therefore, pharmacopuncture or acupuncture treatment methods in this study are not determined in advance, and are performed according to the clinical judgment of the doctor according to the patient's condition. All intervention methods used in this study will be retrospectively recorded in case report form. Upon completing this pRCT, we will present information regarding the specific acupoints used in the results paper.

Reference

2      Alexander Molassiotis. et al. Acupuncture for Cancer-Related Fatigue in Patients With Breast Cancer: A Pragmatic Randomized Controlled Trial American Society of Clinical Oncology 30, 4470-4476 (2012).

What model will be used - "traditional medicine syndrome differentiation" is a very general description.

How will points be selected? According only to traditional theory, or based on the results of previous published research as well? And will the herbs be adjusted according to syndrome as well? For some patients, the combination chosen may be less appropriate than for others.

Response

We will use the validated Korean version of the syndrome differentiation questionnaire (cold and fever, feeble and solid) for syndrome differentiation research. We want to investigate the syndrome differentiation distribution and the correlation of the type of pain by the syndrome differentiation type in Parkinson's disease exploratory. Unfortunately, there is no evaluation of herbal medicine in this study, so we are considering investigation in a follow-up study based on the results.

Will it be possible to analyse results for subgroups of patients with different syndromes or severity/type of PD? Results may be excellent for one, poor for another. Useful findings may be missed without subgroup analysis - although I realise this may mean that more patients need to be recruited.  

Response

We believe that there is a need for subgroup analysis. However, this pilot study is expected to be difficult due to the small number of participants. A large-scale clinical trial is being considered in the follow-up study if a significant clue is identified in this pilot study, although statistically indeterminable.

10 years of clinical experience may mean very little actual experience with PD, unless the senior Korean medicine doctor who is allocating treatment has been working in a neurology setting. I would be happier if I knew more about this senior Korean medicine doctor, and would prefer it if there were case discussions to plan treatment rather than depending on one person's view.

Response

A clinician with ten years of clinical experience is a clinical professor of Korean internal medicine and cardiology and specializes in patients with Parkinson's disease using acupuncture, bee venom, and pharmacoacupuncture. In addition, he has published several papers in SCIE-level journals on the effectiveness of acupuncture in Parkinson's disease.

Author Response

RE: ijerph-2068722, Effectiveness and therapeutic mechanism of pharmacopuncture for pain in Parkinson's disease: a study protocol for a pilot pragmatic randomized, assessor-blinded, usual care-controlled, three-arm parallel trial

Dear Editor and Reviewers,

We appreciate the Reviewers' helpful comments and suggestions. We have added more detailed information to address the issues raised and provided further explanations. In addition, the overall English grammar was corrected, and some contents were modified according to the journal guidelines. The responses to the Reviewers' comments are presented below.

Response to the Reviewer #3’s Comments

 Response

We have mentioned this point in section 1 section of the revised manuscript as follows: “As such, PA is used to treat pain symptom and for nerve recovery. The pain in PD is caused by mostly musculoskeletal pain activated by mechanical, thermal, or mechanical stimuli related to nonneural tissues and related to increased muscle rigidity. And PD pain is also classified as neuropathic pain associated to a lesion or disease of the peripheral or central somatosensory system[4]. therefore, PA will help relieve musculoskeletal pain and neuropathic pain in PD.’

Response

We have mentioned this point in section 2.5 of the revised manuscript as follows: “acupoints centered on the area where the participant complains of pain”.

Pragmatic randomized controlled trial (pRCT) design focusing on effectiveness rather than efficacy that best reflects the likely clinical response in practice.2 Therefore, pharmacopuncture or acupuncture treatment methods in this study are not determined in advance, and are performed according to the clinical judgment of the doctor according to the patient's condition. All intervention methods used in this study will be retrospectively recorded in case report form. Upon completing this pRCT, we will present information regarding the specific acupoints used in the results paper.

Reference

2      Alexander Molassiotis. et al. Acupuncture for Cancer-Related Fatigue in Patients With Breast Cancer: A Pragmatic Randomized Controlled Trial American Society of Clinical Oncology 30, 4470-4476 (2012).

Response

We evaluate adverse reactions as follows. "The causal link between AEs and the procedure will be categorized into six stages as determined by the research clinician: Definitely related, Probably related, Possibly related, Probably not related, Definitely not related, and Unknown"

Round 2

Reviewer 3 Report

This research would be an pioneer in PD and pharmacoacupuncture. Protocol is quit complete after revised. We are all looking forward to the research result of your team.